# Median Nerve Stimulation for Treatment of Tics: Randomized, Controlled, Crossover Trial

**DOI:** 10.3390/jcm12072514

**Published:** 2023-03-27

**Authors:** Ann M. Iverson, Amanda L. Arbuckle, Keisuke Ueda, David Y. Song, Emily C. Bihun, Jonathan M. Koller, Michael Wallendorf, Kevin J. Black

**Affiliations:** 1Washington University School of Medicine, Washington University in St. Louis, St. Louis, MO 63110, USA; 2Department of Psychiatry, Washington University in St. Louis, St. Louis, MO 63110, USA; 3Department of Neurology, Washington University in St. Louis, St. Louis, MO 63110, USA; 4University of Rochester School of Medicine and Dentistry, Rochester, NY 14642, USA; 5Division of Biostatistics, Institute for Informatics, Washington University in St. Louis, St. Louis, MO 63110, USA; 6Departments of Psychiatry, Neurology, Radiology, and Neuroscience, Washington University in St. Louis, St. Louis, MO 63110, USA

**Keywords:** Tourette syndrome/therapy, tic disorders/therapy, transcutaneous electric nerve stimulation, randomized controlled trial, crossover studies

## Abstract

A prior study showed that rhythmic, but not arrhythmic, 12 Hz stimulation of the median nerve (MNS) entrained the sensorimotor cortex EEG signal and found that 10 Hz MNS improved tics in Tourette syndrome (TS). However, no control condition was tested, and stimulation blocks lasted only 1 min. We set out to replicate the TS results and to test whether tic improvement occurs by the proposed cortical entrainment mechanism. Preregistration was completed at ClinicalTrials.gov, under number NCT04731714. Thirty-two people with TS, age 15–64, completed two study visits with repeated MNS on and off blocks in random order, one visit for rhythmic and one for arrhythmic MNS. Subjects and staff were blind to order; a video rater was additionally blind to stimulation and to the order of visits and blocks. Rhythmic MNS at 10 Hz improved tics. Both rhythmic and arrhythmic 12 Hz MNS improved tic frequency, intensity, and urges, but the two treatments did not differ significantly. Participant masking was effective, and there was no carryover effect. Several participants described a dramatic benefit. Discomfort was minimal. There was no evidence that the MNS benefit persisted after stimulation ended. These results replicate the tic benefit from MNS but show that the EEG entrainment hypothesis cannot explain that benefit. Another electrophysiological mechanism may explain the benefit; alternatively, these data do not exclude a placebo effect.

## 1. Introduction

Tourette syndrome (TS) and other chronic tic disorders (CTD) are associated with a reduced quality of life, and current treatment options do not fully meet patient needs [1]. Medications are no more than 50–60% effective in randomized controlled trials, and patients often choose to discontinue medication treatments due to associated side effects [2]. Behavioral therapies require trained therapists who are familiar with tics, and patients must have the ability to attend regularly and to actively participate [3]. However, weekly visits to psychologists are impractical for many patients, especially for those in rural areas or with limited resources [4]. Patients desire new treatment options to improve their quality of life [5].

Morera Maiquez et al. from the University of Nottingham recently proposed peripheral electrical stimulation of the median nerve (MNS) as a treatment for TS and other CTD [6]. The rationale included repetitive transcranial magnetic stimulation (TMS) treatment studies of TS and a study showing that rhythmic, but not arrhythmic, transcranial magnetic stimulation (TMS) pulses entrained cortical oscillations [7]. They hypothesized that rhythmic peripheral nerve stimulation, which, in distinction to TMS, could be delivered portably, might also entrain cortical oscillations and improve tics if delivered at a frequency previously associated with decreases in motor activity. They chose the mu frequency range (8–14 Hz) due to prior associations with motor function.

The Nottingham group demonstrated that 12 Hz rhythmic stimulation of the median nerve evoked synchronous contralateral EEG activity over the primary sensorimotor cortex. The stimulation created small but statistically significant dampening of the initiation of voluntary movements, but did not cause meaningful distraction, as measured by performance on a cognitively demanding test. A follow-up study found similar effects of MNS on cortical activity using magnetoencephalography [8]. In 19 TS patients treated with 10 Hz MNS, blinded video ratings showed a significant reduction in tic counts and tic severity during stimulation. Additionally, patients reported a reduced urge to tic during MNS. Some participants reported benefits lasting after MNS was turned off. Though the results of this study suggest MNS as a promising potential treatment option, the TS experiment had no active control treatment.

However, that study suggests an optimal control condition. Arrhythmic MNS did not evoke EEG activity as rhythmic stimulation did, nor did it decrease volitional movements in healthy volunteers. Participants could not distinguish between rhythmic and arrhythmic stimulation at the same mean frequency. These results make arrhythmic stimulation an ideal active control condition to test the proposed mechanism of MNS for tics and possibly to exclude a placebo effect.

We designed the present study to replicate the results of the Nottingham study, to test the proposed mechanism of action using arrhythmic MNS as a control arm, and to systematically examine the duration of treatment benefit after stimulation ends. 

## 2. Materials and Methods

### 2.1. Participants

Participants were recruited through clinical referrals, referrals from the University of Nottingham research team, advertising, and word of mouth. Recruitment and study completion occurred between June 2021 and April 2022. Authors A.L.A., E.C.B., or K.J.B. enrolled participants.

Inclusion criteria for all subjects were: age 15–64 inclusive at initial screening visit, current DSM-5 Tourette’s Disorder or Persistent (Chronic) Tic Disorder, and at least 1 tic per minute (on average) during the 5-min baseline video session on the first visit (as scored during the session by the investigator). Exclusion criteria were: unable to complete study procedures for any reason, having an implanted device that could be affected by electrical current, pregnancy known to participant or (for children) to the parent, known or suspected primary genetic syndrome (e.g., Down syndrome, Fragile X), intellectual disability (known, or likely from history and examination), head trauma with loss of consciousness for more than 5 min, significant neurologic disease (exceptions included febrile seizures or uncomplicated migraine), severe or unstable systemic illness, factors (such as exaggerated signs) that in the judgement of the principal investigator made the video recording or YGTSS an inaccurate assessment of tic severity, judged to be unlikely to complete study procedures or to return for later visits, change in somatic or psychotherapeutic treatment in the 2 weeks preceding the first stimulation visit, or planned change in somatic or psychotherapeutic treatment between the 2 stimulation visits.

A sample size of 32 participants was chosen, estimated to provide 75% power to detect clinically meaningful (25%) improvement in Yale Global Tic Severity Scale total tic scores (YGTSS TTS) [9], based on the mean response and variance in the original Nottingham study [6].

### 2.2. Trial Design

The study protocol was pre-registered prior to enrolling the first participant and can be consulted for additional methodological details [10]. The study was a randomized, double-blind, crossover design with two study visits separated by a washout period of one week. Study visits occurred at Washington University in St. Louis School of Medicine. A crossover design was chosen to maximize power, since carryover effect was expected to be negligible. Makeup sessions were allowed for technical or other problems, which occurred for 4 participants.

The two stimulation sessions were identical, except that one session used rhythmic median nerve stimulation, and the other used arrhythmic stimulation. Arrhythmic simulation used the same mean frequency, i.e., the same number of total pulses per minute, but the arrhythmic stimulation used a random inter-pulse interval. Session order was randomized using simple randomization via random.org by author K.J.B. prior to enrolling the first participant, with half of the participants receiving rhythmic stimulation first, and half of the participants receiving arrhythmic stimulation first. Author J.M.K., who was not present at the study visits, used the randomization sequence to program the stimulator. Participants and study staff present at the visit did not know the order of rhythmic and arrhythmic stimulation. Sessions were video recorded for later analysis by a blinded independent rater.

Prior to the first session, participants answered questions about family and medical history, treatment history, tic symptom history, current symptom status, and other comorbid symptom status. A clinician reviewed this information with the participants at the first visit and performed a neurological and psychiatric examination as appropriate. Participants were told that stimulation would feel similar on the two days, but that on one day, the stimulation would be given in a manner we expected would be effective, and on the other day, in a manner we expected would be ineffective, but that we did not know whether either or both would be effective. 

A 5-min baseline tic video was recorded and scored prior to the stimulation protocol to ensure the participant met the inclusion criterion of tics at a rate of at least one tic per minute. The stimulation protocol is described below. Prior to the second visit, participants updated health history, treatment, current tic symptoms, and other comorbid symptom status. On the second visit, neurologic and psychiatric examinations were repeated as appropriate, and a second 5-min baseline video was recorded prior to the stimulation protocol.

### 2.3. Clinical Assessment Measures

In addition to the YGTSS, assessments at baseline included the Diagnostic Confidence Index (DCI), a measure of classic features of Tourette syndrome [11], demographics, medical history, and neuropsychiatric examination, and the following self-report symptom severity measures: the Adult Tic Questionnaire (ATQ) [12]; the Premonitory Urge for Tics Scale (PUTS) [13]; the ADHD Rating Scale [14], a self-report version [15] of the Yale–Brown Obsessive Compulsive Scale (Y-BOCS) Scale [16,17], the Social Responsiveness Scale (SRS-2), a validated self-report measure of current autistic trait severity [18], and the Edinburgh Handedness Inventory [19]. At the second visit, these assessments were repeated or reviewed.

### 2.4. Stimulation Protocol

A bar electrode with contacts 30 mm apart, center to center, was applied over the median nerve at the right wrist using conductive gel. The anode was distal. The stimulation threshold for movement of the thumb was determined for a train of 8 pulses at 12 Hz, with a pulse width of 200 µs. Stimulation began at 2 mA and gradually increased until a twitch of the thumb was seen; this current was used for all remaining stimulation blocks on that study visit. 

Stimulation was delivered by an SI-200 Isolated Stimulator device (iWorx, Dover, NH (https://web.archive.org/web/20230125235627/, https://iworx.com/products/stimulators/si-200-stimulus-isolator/, accessed on 25 January 2023)), which is designed to limit maximum delivered current to 20 mA. The timing of the stimulation was precisely controlled by a Mega 2560 microcomputer (Arduino, Turin, Italy (https://web.archive.org/web/20230125235755/, https://docs.arduino.cc/hardware/mega-2560, accessed on 25 January 2023)) delivering TTL pulses to the stimulator. Arrhythmic stimulation used a set 1-min sequence of 600 (for 10 Hz) or 720 (for 12 Hz) 200 µs pulses with randomly generated start-to-start pulse intervals between 10 and 200 ms for 10 Hz and between 10 and 164 ms for 12 Hz. Python and Arduino code and the arrhythmic stimulation parameters are available at https://github.com/BlackHershey/MNS/releases/tag/v1.0.0 (accessed on 25 January 2023).

At this point, video recording began, and clocks were synchronized for video and stimulation. Participants had a portable microphone placed near the neck to better detect phonic tics, and the right hand and wrist were positioned out of view of the camera to prevent accidental unblinding of the video rater by MNS-induced hand or finger movements. MNS stimulation was then conducted as follows (see Figure 1): two 1-min blocks of MNS on and two 1-min blocks of MNS off (order randomized), followed by four alternating 5-min blocks (MNS off, on, off, on), followed by from one to four 5-min MNS-off blocks until tics returned to baseline frequency (as judged by both the investigator and independently by the participant). Participants rated their urge to tic just prior to the conclusion of each block. The 1- and 5-min blocks after baseline were conducted continuously without breaks.

### 2.5. Video Analysis

A movement disorders child neurologist (author K.U.) blind to treatment order (rhythmic MNS first or second), visit number (1 or 2), stimulation condition (on vs. off), and order (of the 1-min or 5-min blocks) used a simplified version of TicTimer Web software [20,21] to mark each occurrence of any tic and a REDCap form to provide a maximal rating of motor and phonic tic severity, for each of the 5-min blocks [22,23]. For the 1-min blocks, he used Datavyu software to mark the occurrence of and provide a rating of the severity of each instance of any tic; motor and phonic tics were tracked separately [24].

### 2.6. Outcome Measures

Primary and secondary outcome measures were prespecified in the registered protocol [10]. Primary outcome measures comprised a replication of the Nottingham results (change in tic frequency and severity with 1-min 10 Hz rhythmic MNS on vs. off) and a test of the proposed mechanism (change in tic frequency and severity during rhythmic vs. arrhythmic 12 Hz MNS on and off). Secondary outcome measures included visual analog scale (VAS) ratings of the urge to tic over the past minute, rated just prior to the end of each 5-min block, a test of duration of the benefit (tic frequency in each 1-min bin after the last block of active MNS), and the following at the end of the visit: Clinical Global Impression of Improvement (CGI I) ratings by participant and investigator, rating of discomfort using the CGI Efficacy Index (it was felt to be superior to the VAS measure in the registered protocol) [25], and participant and investigator guess as to whether the visit used active or control stimulation. Any additional outcomes are exploratory.

### 2.7. Statistical Analysis

Outliers were determined at the level of the individual variables before the main analysis. The primary outcome measures were analyzed as follows. Conservatively, two-tailed p values are given unless otherwise stated, even though most hypotheses tested were directional. 

### 2.8. Replication (1-Min Blocks of Rhythmic MNS)

Each occurrence of every tic during the last 40 s of on vs. off 1-min stimulation epochs on the rhythmic MNS day was scored by the blinded rater using the intensity item from the Yale Global Tic Severity Scale (YGTSS), which uses integer scores from 0 (no tics) to 5 (severe). The number of tics (tic frequency) and the mean of the intensity ratings in each block (tic intensity) comprised the dependent variables. This sample design replicated that of Study 3 from Morera Maiquez et al. (2020) [6], though our analysis differed. We log transformed tic count to stabilize variance. We analyzed log tic count and mean tic intensity with a random effect repeated-measures model in which the subject was random, and the correlation within subjects was modeled with a compound symmetry covariance structure. Fixed effects were baseline measure (Block 0), block (Block 1, 2, 3, or 4), stimulation (on, off), stimulation lag (1 for blocks preceded by a block with active stimulation and 0 otherwise), and stimulation lag by stimulation interaction. The lag variable was added to the protocol-specified analysis to test for the persistence of an effect in consecutive blocks.

### 2.9. Randomized Crossover Trial (5-Min Blocks of Rhythmic vs. Arrhythmic MNS)

The dependent variables for the 5-min blocks in the crossover RCT were the number of tics in the block (tic frequency), the greater of the motor tic intensity rating and the phonic tic intensity rating, each rated once for the entire 5-min block (tic intensity), and the strength of the urge to tic rated by visual analog scale just before the end of each block (tic urge). For the crossover design, we analyzed each of these 3 variables using a mixed random effect crossover repeated-measures model (mixed procedure, SAS 9.4, Cary N.C.) in which the subject was a random effect, and within-subject correlation was modeled with a first-order autoregressive covariance structure. Fixed effects were baseline measure (Block 0), sequence (rhythmic or arrhythmic at first visit), treatment (rhythmic or arrhythmic), visit (first or second), block (5-min blocks numbers 5, 6, 7, 8, or 9), and treatment by block interaction. A non-significant sequence indicated that carryover between visits was negligible, such that both visits could be analyzed. We tested stimulation by comparing the mean of blocks with stimulation to those without. The pre-study protocol had specified a simpler repeated-measures ANOVA analysis with effects of visit, sequence, and stimulation on vs. off. The analysis used was adopted, as it better accounted for the study design.

### 2.10. Duration of Benefit

We tested whether tic frequency changed over the 5–20 min following the end of stimulation. Since the duration of observation was longer in participants whose tics did not appear to have returned to baseline frequency—i.e., since data from Blocks 10–12 were not missing at random—we used a last observation carried forward (LOCF) analysis, as follows. For participants with <20 min of post-MNS observation, the mean tic rate (tics/min) in the last observed 5-min OFF block was carried forward for each minute through post-stimulation minute 20. As pre-registered, we compared post-MNS tic rate to tic frequency in Block 8, the last 5-min stimulation block (testing for a change in tic rate after stimulation ended). We also performed the analysis using as baseline Block 7, the last stimulation-off block before the last stimulation block, as on further reflection, it was felt this better tested for a change in tic rate induced by the last ON block. In both cases, since tic counts were non-normal, we used the nonparametric Friedman test.

### 2.11. Additional Analyses

Success of blinding was assessed using the binomial distribution. For all the above, a *p* value < 0.05 was accepted as significant. Descriptive statistics were used to summarize data in Table 1 and CGI scores, with a t-test or χ^2^ test, as appropriate. Rater agreement used the Spearman correlation test, since the underlying data were non-normal.

To test whether any individual characteristics predicted a differential response to rhythmic or arrhythmic stimulation, participants were assigned a score of their arrhythmic improvement subtracted from their rhythmic improvement. Improvement on each day was measured as the difference between the average number of tics in blocks with stimulation off and blocks with stimulation on. Therefore, the difference score was positive for those participants who improved more on the rhythmic day and negative for those who improved more on the arrhythmic day. This difference score was compared against individual baseline characteristics using Spearman correlation for continuous variables and a Mann–Whitney U test for categorical variables.

For comparisons of stimulation thresholds, descriptive statistics and paired t-tests were used to compare visit days and types of stimulation.

## 3. Results

A CONSORT checklist adapted for randomized crossover trials is attached in the Appendix A [26].

### 3.1. Participants

Thirty-two participants enrolled in this study and completed both visit 1 and visit 2 (Figure 2). Half had rhythmic MNS on the first visit, and the other half arrhythmic. One participant ended one visit early due to discomfort after the first 5-min active MNS block on the arrhythmic day; all other visits were completed in full, requiring a makeup visit in four cases. One participant’s two visits occurred only 1 day apart; all others’ two visits were separated by at least a week. The video recordings after baseline were lost from one participant’s arrhythmic MNS visit, so n = 31 for analyses of tic frequency and intensity with arrhythmic stimulation. 

Participants spanned the allowed age range from 15 to 64 years. They had a typical tic history for patients at a referral center (DCI score mean = 60) and a wide range of current tic severities, on average moderately severe (mean YGTSS TTS = 26; see Table 1). No changes in treatment were reported between sessions for any of the participants.

### 3.2. Replicating the University of Nottingham TS study

Both the number of tics (least squares (l.s.) mean tic count 12.1 off, 8.9 on; stimulation effect *p* = 0.01) and the intensity ratings for tics (l.s. mean tic count 2.5 off, 2.3 on; *p* = 0.079) decreased during 1-min 10 Hz rhythmic stimulation. The decrease in tic intensity was statistically significant only if assessed with a one-tailed test (*p* = 0.04). 

### 3.3. Testing the Hypothesized Electrophysiological Mechanism of Benefit

#### 3.3.1. Tic Frequency

Using the 12 Hz MNS frequency from the EEG experiment from Morera Maiquez et al., there was no evidence of a between-visit carryover effect on tics (sequence *p* = 0.42). On average, participants had 28% fewer tics on visit 2 than on visit 1 (l.s. difference 22.5, 95% confidence interval (CI) 8.3–36.8; *p* = 0.0022). There were 19% fewer tics on average when stimulation was on versus off (l.s. difference 14.4, 95% CI 9.1–19.6; block effect *p* < 0.0001). However, arrhythmic and rhythmic stimulation had similar effects (l.s. estimate of differential treatment effect 0.96, 95% CI (−12.9, 14.8), treatment effect *p* = 0.89, treatment × block *p* = 0.92; see Figure 3).

#### 3.3.2. Tic Intensity

There was no evidence of a carryover effect (sequence *p* = 0.41). Tic intensity was similar on the first and second visits (*p* = 0.94). Tic intensity was significantly though minimally affected by stimulation (l.s. means 3.23 off, 3.11 on, difference 0.12, 95% CI 0.01–0.23; *p* = 0.02), but arrhythmic and rhythmic stimulation had similar effects (l.s. estimate of differential treatment effect 0.03, 95% CI (−0.19, 0.26), treatment effect *p* = 0.77, treatment × block *p* = 0.80; see Figure 4).

#### 3.3.3. Urge to Tic

There was no evidence of a carryover effect on urge to tic (sequence *p* = 0.27), nor was there a difference in urge to tic between visit 1 and visit 2 (*p* = 0.59). Urge to tic improved during stimulation (l.s. difference 10.8, 95% CI 7.9–13.7; *p* < 0.0001), but there was no difference between arrhythmic and rhythmic stimulation (l.s. estimate of differential treatment effect −0.6, 95% CI (−6.1, 4.8), treatment *p* = 0.82, treatment × block *p* = 0.88) (see Figure 5).

### 3.4. Duration of Benefit after the End of Stimulation

In the 1-min blocks, the stimulation lag variable was not significant for either analysis (all ps > 0.2), suggesting no carryover for more than 20 s after stimulation. More directly, there was no significant change in tic frequency comparing the 20 min after to the 5 min before the last MNS ON block (Block 7, rhythmic *p* = 0.684, arrhythmic *p* = 0.356). (Compared to the last stimulation block, Block 8, rhythmic *p* = 0.314, arrhythmic *p* = 0.239.) Figure 6 shows median tic frequency before, during, and after the last active stimulation block (see also Appendix A).

### 3.5. Clinical Response and Harms

Most participants reported a substantial benefit from stimulation: 25 of 32 rated at least one visit as “much improved” or “very much improved” on the CGI-I. Investigator ratings were similar (26 of 32). Mean ± SD CGI-I participant ratings of response were 2.69 ± 1.03 (rhythmic) and 2.39 ± 0.99 (arrhythmic), where 2 = much improved and 3 = minimally improved. CGI-I investigator ratings were similar, 2.41 ± 0.91 (R) and 2.56 ± 1.01 (A).

There were no serious adverse events. In fact, stimulation was well tolerated by participants, with discomfort rated as none or minimal on 57 of the 64 visits. Further details from the CGI–Efficacy Index are shown in the Appendix A. 

### 3.6. Free-Text Responses

Some patients experienced a dramatic benefit from the stimulation (see Appendix A). Comments from four participants who rated tics on the rhythmic MNS day as “very much improved” included “palpable relief from my need to tic … first time in 50 years I did not feel the urge to tic,” a “calm, … almost a little euphoric feeling,” “a sense of just peace, or … not sure how to describe it, maybe what people without Tourette’s feel like … no urge, or almost,” and “no tics at all when it [MNS] was on.” However, this dramatic benefit was not exclusive to rhythmic stimulation. Three participants who self-rated as “much improved” on the arrhythmic MNS day reported feeling “more at ease, less feelings of jittery,” “less urge to tic,” and “urge [to tic] decreased.” The Appendix A contains further participant comments on the stimulation. 

### 3.7. Participants’ Future Treatment Plans

Many participants were interested in the possibility of a portable device that would deliver stimulation similar to the stimulation they had just received (25 of 32 on the rhythmic day, 28 of 32 arrhythmic). Not surprisingly, 31 of the 32 participants who completed this study agreed to take part in an open-label follow-up study, which provided them with their own TENS device to take home and use [27].

### 3.8. Adequacy of Blinding

Participants guessed no better than chance whether they were receiving the expected efficacious treatment or the control treatment (30 of 64 correct, binomial *p* = 0.73). The investigator (K.J.B.) guessed 40 of 64 instances correctly (*p* = 0.03). A separate author (K.U.) performed the blinded ratings for tic frequency and tic intensity. 

### 3.9. Individual Characteristics Predicting Improvement with MNS

We compared responders (investigator CGI-I rating of 1 or 2 on rhythmic stimulation day) to non-responders on the following characteristics: age, sex, history of complex motor or vocal tics, voluntary tic suppression, and coprophenomena, YGTSS TTS, PUTS score, SRS total T score, and ADHD total score and subscores. None differed significantly between groups, though complex vocal tics were more common in responders (*p* = 0.075). 

### 3.10. Other Exploratory Analyses

We tested whether any of these individual characteristics predicted a differential response to rhythmic or arrhythmic stimulation (see Methods). Only the ADHD hyperactivity-impulsiveness subscore correlated significantly with participants’ improvement difference score (rho = −0.45, *p* = 0.01), in that those with greater hyperactivity-impulsiveness were more likely to improve on the arrhythmic day.

Eight participants rated improvement R > A (rhythmic better than arrhythmic), and 15 rated A > R. The investigator rated 12 as R > A and 11 as A < R. Tic number in the blind tic counts decreased more on the rhythmic day in 12 participants and on the arrhythmic day in 18.

Tic counts for the first visit’s baseline session from the investigator, “live,” and from the blinded rater correlated significantly (Spearman’s rho = 0.66, *p* < 0.0001) (Figure 7).

The stimulation threshold was similar on the two visits (*p* = 0.37), median 7.1 mA (IQR 5.8–8.0, range 3.2–12.0) on visit 1 and 6.6 (5.9–8.45, range 4.2–15.6) on visit 2. The stimulation threshold was also similar for both types of stimulation (*p* = 0.21), median 7.1 mA (IQR 5.55–8.15, range 3.2–15.6) for arrhythmic stimulation and 6.6 mA (IQR 6.0–8.2, range 4.0–12.4) for rhythmic stimulation.

## 4. Discussion

This study is the first controlled trial to test median nerve stimulation as a treatment for tics. We report two main conclusions. Our first aim was to replicate the results of Morera Maiquez et al. using identical stimulation and assessments. We confirm that 1 min of 10 Hz rhythmic MNS improved tic frequency and intensity compared to 1-min MNS-off blocks. However, we did not find evidence to support their suggestion that benefit might last after MNS is turned off; this information is crucial for designing chronic treatment.

We next tested the 12 Hz stimulation from the Nottingham EEG experiment. Tic frequency and intensity by blinded ratings and self-reported urge to tic all improved significantly during 5-min active stimulation blocks. However, they improved similarly with arrhythmic stimulation, which does not entrain cortical EEG. Therefore, contralateral somatomotor cortex power and coherence in the mu frequency range does not mediate the improvement in tics. 

We were not able to disprove a placebo explanation for the benefit. Our study, after all, provided a somewhat dramatic form of treatment, which many of the participants expected might be effective, having seen news coverage of the 2020 report. Nevertheless, the improvement in tics with MNS was clinically substantial—in fact, remarkable in several participants. MNS may have true efficacy, given the severity and chronicity of TS in this sample and the number of past treatments they had tried without meaningful benefit, usually including treatments known to be efficacious and thus likely carrying similar expectation of benefit. MNS did not create substantial distraction from a demanding cognitive task, arguing against one possible mechanism for nonspecific benefit [6]. 

One may hypothesize that arrhythmic stimulation exerts a benefit on tics but does so via a different electrophysiological mechanism than entraining the sensorimotor cortex at a frequency associated with decreased voluntary movement. Given the current state of our knowledge, we can only speculate. One possibility is that tics reflect pathological firing patterns in relevant brain networks, and that MNS disrupts this abnormal firing pattern. Such a mechanism has been hypothesized to explain the beneficial effects of deep brain stimulation for movement disorders [28]. However, given the present results, this disruption in the case of MNS for TS could depend on the mean frequency of the stimulation or on its mere presence, but not on its rhythmicity. Another possibility is suggested by the observations that an urge to tic is nearly universal in TS, that a third to a half of people with TS have hyperactivity, that physical exercise tends to improve tics, and that sensory hypersensitivity is very common in TS. These observations suggest a hypothesis that the TS brain expects a particular kind and amount of input to the primary somatosensory cortex (S_1_), and that the somatosensory feedback generated by tics or intentional movements can provide that input when it is not supplied by the environment. In this scenario, the input MNS provides to S_1_ may decrease both the urge to tic and tic severity, as we observed with rhythmic or arrhythmic MNS. Other electrophysiological mechanisms that could explain a specific tic reduction benefit with arrhythmic MNS are no doubt possible. For any such mechanism, a control condition other than arrhythmic stimulation may reveal differential efficacy for the active treatment arm, demonstrating a benefit not explained by a placebo effect.

Alternatively, rhythmic and arrhythmic MNS may benefit different subgroups of participants. In fact, participants with greater self-rated hyperactivity-inattentiveness responded more favorably to arrhythmic than to rhythmic stimulation. However, this result was from an exploratory analysis, uncorrected for the number of variables examined.

We also show that MNS was well tolerated, and all participants but one joined a follow-on open-label study with a portable stimulator, the results of which are reported separately [27]. 

### 4.1. Limitations

Crossover studies are potentially subject to carryover effects. However, the primary outcome measures showed no evidence of such effects, consistent with the 1-week separation of the two visits, and no evidence that the benefit from MNS lasted even minutes after stimulation ended. Second, the investigator guessed the type of stimulation correctly statistically more often than chance would explain, though with only 62% accuracy. Fortunately, participants remained blinded (guessing correctly 47% of the time), and none of the key outcome measures were influenced by the investigator. Third, we did not test other variations on MNS, so we cannot address the specificity of the median nerve (vs. any other peripheral nerve), the specificity of 10–12 Hz as the stimulation frequency, or whether there is a lateral preference for MNS benefit. Fourth, most of our participants found the study via social media; this selection could theoretically affect generalizability to clinical populations. However, our participants were typical for clinical populations in terms of disease features and course (DCI score) and had a relatively high current severity (YGTSS scores), including one participant with a YGTSS score of 100 (the maximum possible score).

### 4.2. Future Directions and Conclusion

We are interested in testing some of the unanswered questions listed in the previous paragraph and investigating the mechanism by which MNS improves the severity of tics and the urge to tic. The Nottingham group is performing a RCT of subchronic MNS using a portable wristwatch-style pulse generator in TS (https://web.archive.org/web/20230126030424/, accessed on 26 January 2023, https://www.neupulse.co.uk/median-nerve-stimulation-on-tourette-syndrome-trial/, accessed on 26 January 2023). Informed by preliminary results from the present study, they are using a sub-motor-threshold stimulus as the control condition. 

In summary, MNS is well tolerated and reduces tics, but not via the initially proposed mechanism of increased EEG power and coherence at 10–12 Hz over the contralateral motor cortex.

## Figures and Tables

**Figure 1 jcm-12-02514-f001:**
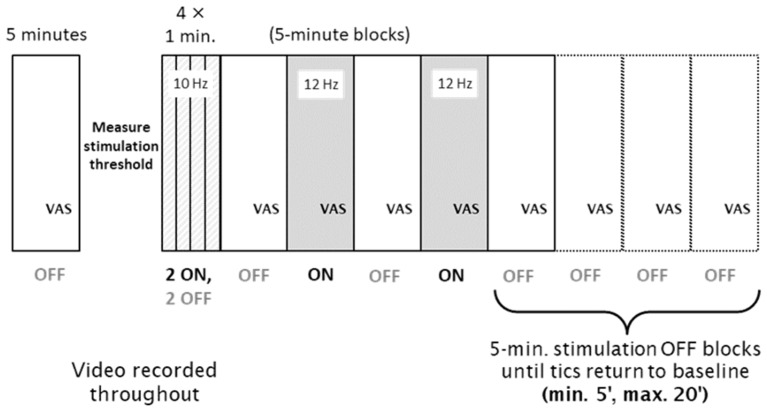
Diagram of study procedures during each visit. A baseline 5 min Block 0 was followed by measurement of the stimulation threshold, then by 1 min Blocks 1-4, and finally by 5 min Blocks 5-12. Stimulation was ON in two of the four 1 min blocks (selected at random for each subject) and in Blocks 6 and 8 and was OFF in all other blocks.

**Figure 2 jcm-12-02514-f002:**
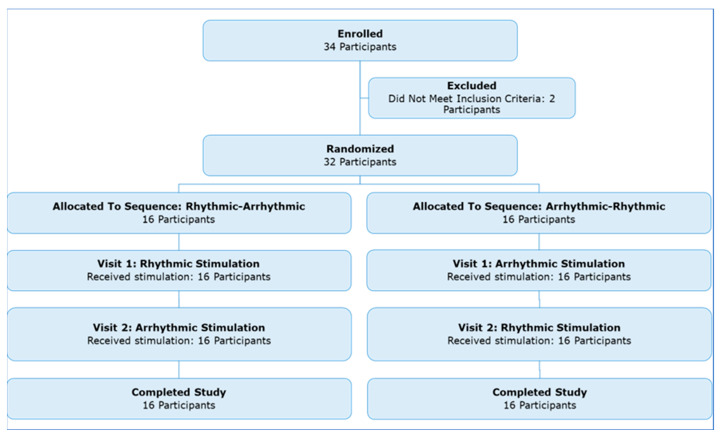
Participant enrollment flowchart.

**Figure 3 jcm-12-02514-f003:**
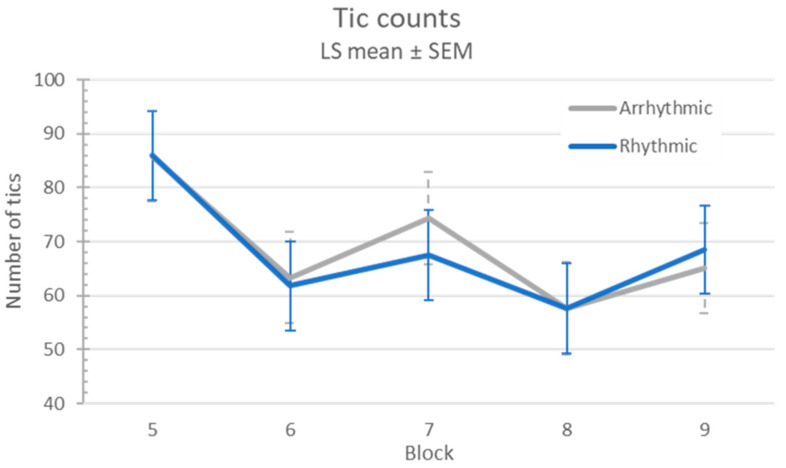
Tic frequency response to MNS.

**Figure 4 jcm-12-02514-f004:**
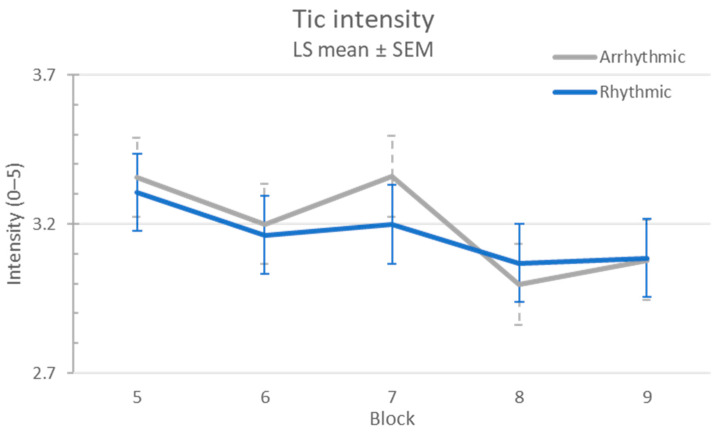
Tic intensity response to MNS.

**Figure 5 jcm-12-02514-f005:**
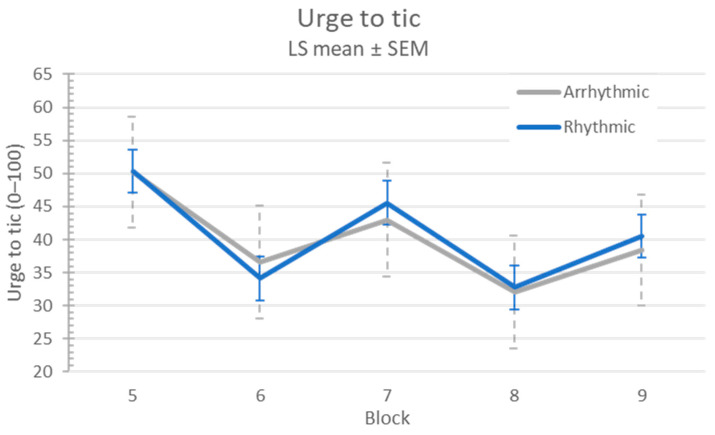
Tic urge response to MNS.

**Figure 6 jcm-12-02514-f006:**
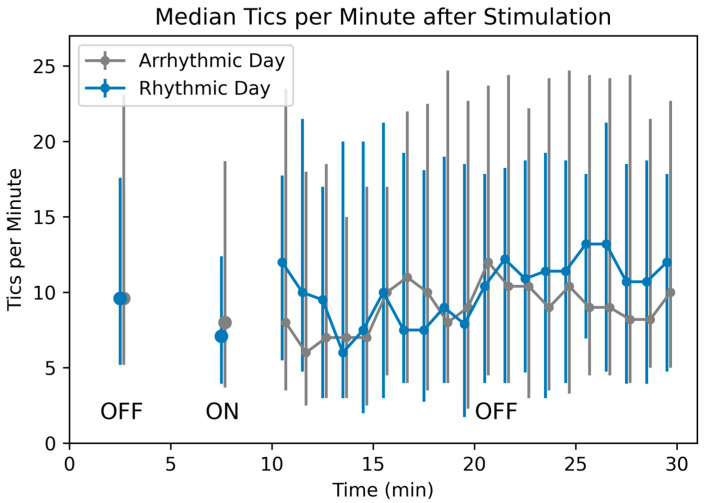
Tic frequency after stimulation ends. Median tic frequency in Block 7 (the OFF block before the last active 5 min stimulation block) is plotted at time = 2.5 min, median tic frequency in Block 8 (the last ON block) at 7.5 min, and median tic frequency for each of the 20 subsequent minutes with stimulation OFF is plotted at times 10.5–29.5 min. The last 15 min include LOCF data, as described in Methods. Vertical bars represent 25th and 75th percentiles.

**Figure 7 jcm-12-02514-f007:**
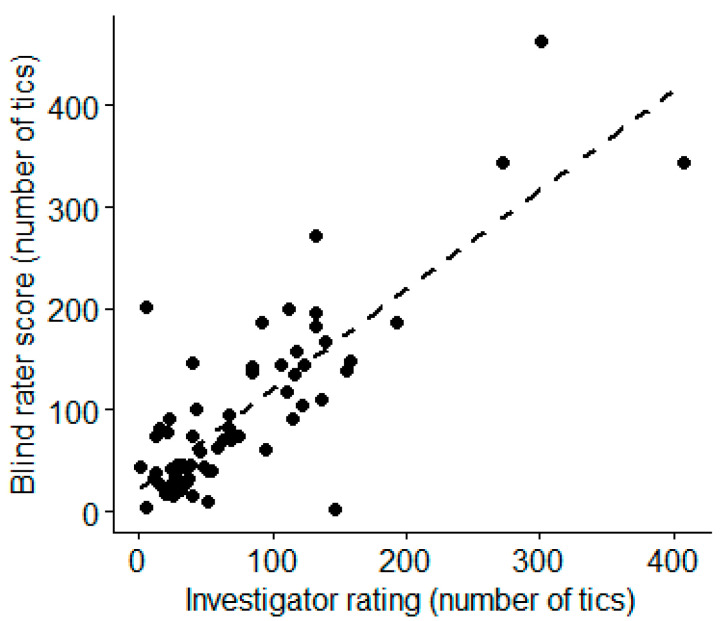
Correlation between investigator and blind rater rating of baseline tic frequency. (Spearman’s rho = 0.66, *p* < 0.0001).

**Table 1 jcm-12-02514-t001:** Participant characteristics.

Characteristic	Arrhythmic then Rhythmic Stimulation (n = 16)	Rhythmic then Arrhythmic Stimulation (n = 16)	Total (n = 32)
Age (years, mean ± SD)	31.25 ± 16.16	36.81 ± 16.81	34.03 ± 16.51
Sex	9M, 7F	12M, 4F	21M, 11F
Handedness (EHI)	16R, 0L	14R, 2L	30R, 2L
Self-reported race:			
White	14	15	29
Black	1	0	1
Asian	0	1	1
More than one race	1	0	1
Hispanic or Latino	1	0	1
Tourette’s Disorder, DSM-5	15	16	31
Persistent Motor Tic Disorder, DSM-5	1	0	1
Family history of tics (first-degree relatives)	7	6	13
DCI score	61.69 ± 17.84	59.13 ± 22.54	60.41 ± 20.04
Marked distress/impairment in a life role, ever	16	16	32
Marked distress/impairment in a life role in the past week, visit 1	9	10	19
Phonic tics (lifetime)	15	15	30
Phonic tics (past week), visit 1	15	15	30
Complex tics (lifetime)	16	11	27
Complex motor	15	11	26
Complex vocal	6	7	13
Complex tics (past week), visit 1	15	15	30
Complex motor	13	15	28
Complex vocal	7	3	10
Coprophenomena, ever	5	4	9
Coprolalia	3	4	7
Copropraxia	3	2	5
Coprophenomena (past week), visit 1	3	2	5
Coprolalia	2	2	4
Copropraxia	1	2	3
Ever sought treatment or diagnosis	16	14	30
Lifetime adequate behavior therapy for tics or OCD	3	0	3
Currently taking medication for tics	4	3	7
YGTSS total tic score	26.38 ± 8.75	25.25 ± 6.37	25.82 ±7.85
YGTSS motor	16.38 ± 3.54	15.69 ± 3.40	16.04 ± 3.47
YGTSS phonic	10.00 ± 6.38	9.56 ± 4.30	9.78 ± 5.55
YGTSS impairment	20.63 ± 18.31	18.44 ± 13.51	19.54 ± 15. 52
ATQ score	40.06 ± 22.56	36.44 ± 20.54	38.25 ± 21.30
PUTS score	20.31 ± 7.53	22.50 ± 7.11	21.41 ± 7.29
Tics per minute before stimulation, visit 1 (live rating)	16.91 ± 14.28	18.81 ± 17.18	17.86 ± 15.57
Tics per minute before stimulation, visit 1 (blind rating)	11.84 ± 9.67	17.34 ± 19.85	14.74 ± 15.58
Tic intensity (YGTSS) before stimulation, visit 1			
Motor	3.06 ± 1.39	3.44 ± 0.89	3.25 ± 1.16
Phonic	0.88 ± 1.50	1.38 ± 1.75	1.13 ± 1.62
Y-BOCS (self-rated)	7.81 ± 6.60	4.75 ± 5.00	6.28 ± 5.96
ADHD score past week (self-rated)	12.75 ± 12.05	13.00 ± 14.35	12.88 ± 13.04
SRS total (T score)	54.69 ± 12.92	51.63 ± 10.07	53.16 ± 11.50

## Data Availability

Individual subject data are available via the Open Science Foundation page for the project: DOI:10.17605/OSF.IO/MTBZF.

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
