# Peer review of "Median Nerve Stimulation for Treatment of Tics: Randomized, Controlled, Crossover Trial"

_jcm, 2023, doi:10.3390/jcm12072514_

Round 1

Reviewer 1 Report

Thank you so much for providing me the opportunity to review this article. I think it is very relevant and well written and it was prepared by the study group with an extensive expertise in this topic. I have only some small suggestions that the authors might consider to introduce. In the methodology section, what is meant by the statement that the participants were included on the basis of the referals from the University of Nottingham - did they flight from England to the US to complete the study? What was the rationale behind the inclusion of participants with at least 1 tic in 5 minutes of video observation? Is it based on some evidence based tic assessment? Would appreciate the clarification. What is the meaning of the exclusion criteria of "exaggerated signs" mentioned in the exclusion criteria? Finally, I think it is important to mention in the discussion what is the potential mechanism behind the efficacy of this therapy?

Author Response

Point 1: In the methodology section, what is meant by the statement that the participants were included on the basis of the referals from the University of Nottingham - did they flight from England to the US to complete the study?

Response 1: This statement refers to potential participants in the United States who saw the news coverage of the initial University of Nottingham study and contacted the research team there to learn more. The University of Nottingham team then referred those participants to our study team.

Point 2: What was the rationale behind the inclusion of participants with at least 1 tic in 5 minutes of video observation? Is it based on some evidence based tic assessment? Would appreciate the clarification.

Response 2: We wanted to ensure that the participants who enrolled in the study had tics that were frequent enough for us to be able to observe any potential improvement in tics due to treatment. For example, if a potential participant had zero tics in a baseline 5 minute block, it would be impossible for them to demonstrate any improvement in their tics. The specific cutoff of at least one tic per minute, averaged over a 5 minute baseline period, was somewhat arbitrary.

Point 3: What is the meaning of the exclusion criteria of "exaggerated signs" mentioned in the exclusion criteria?

Response 3: In this case, ‘exaggerated signs’ was meant to encompass participants who are exhibiting symptoms that are not fully consistent with physiologic tics, such as those participants with functional neurologic disoder or other similar phenomena.

Point 4: Finally, I think it is important to mention in the discussion what is the potential mechanism behind the efficacy of this therapy?

Response 4: In this study, we demonstrate that the mechanism cannot be the one originally hypothesized. We discussed the posibility that the mechanism is a placebo response and we mention very briefly in lines 411-413 the idea of a different electrophysiological mechanism underlying the observed effect: (“One may hypothesize that arrhythmic stimulation exerts benefit on tics but does so via a different electrophysiological mechanism than entraining sensorimotor cortex at a frequency associated with decreased voluntary movement”). At this stage of our knowledge, we would assert that addressing the mechanism further would be pure speculation. However, we have added in some further conceptual information surround possible mechanisms in lines 413-418.

Thank you very much for your suggestions. Please see the attachment.

Reviewer 2 Report

The work presented is of enormous interest and relevance. It is worth paying attention to new treatments to improve tics, which compromise the quality of life of patients suffering from this disorder.

Below we present a number of aspects that could improve the excellent work presented:

- In line 44 the name should be included before the abbreviation (CTD). 

- The aim of the research work needs to be made more specific. It is too vague.

- It would be interesting to include a table with the main socio-demographic characteristics of the participants in the study.

- It is necessary to include a section with the conclusions of the study. This part has not been included in the submitted manuscript. 

Author Response

Point 1: In line 44 the name should be included before the abbreviation (CTD).  

Response 1: The acronym CTD (“chronic tic disoders”) is fully spelled out prior to its first usage in line 33. Therefore, we did not repeat the name again in line 44. 

Point 2: The aim of the research work needs to be made more specific. It is too vague. 

Response 2: We address the aim of our study in lines 68-70 (“We designed the present study to replicate the results of the Nottingham study, to test the proposed mechanism of action using arrhythmic MNS as a control arm, and to systematically examine the duration of treatment benefit after stimulation ends”). 

Point 3: It would be interesting to include a table with the main socio-demographic characteristics of the participants in the study. 

Response 3: We include all socio-demographic information that we collected from our participants (age, sex, race, and ethnicity) in Table 1: Participant Characteristics (line 282). 

Point 4: It is necessary to include a section with the conclusions of the study. This part has not been included in the submitted manuscript. 

Response 4: The conclusions of our study appear initially in lines 391-401 (“We report two main conclusions. Our first aim was to replicate the results of Morera Maiquez et al using identical stimulation and assessments. […] Therefore, contralateral somatomotor cortex power and coherence in the mu frequency range does not mediate the improvement in tics”). Additionally, our conclusions are further stated in section 4.2. Future Directions and Conclusion, specifically lines 461-463 (“In summary, MNS is well tolerated and reduces tics, but not via the initially proposed mechanism of increased EEG power and coherence at 10-12 Hz over contralateral motor cortex”).

Thank you very much for your suggestions. Please see the attachment.
